# Differences in Adipose Gene Expression Profiles between Male and Female Even Reindeer (*Rangifer tarandus*) in Sakha (Yakutia)

**DOI:** 10.3390/genes13091645

**Published:** 2022-09-13

**Authors:** Melak Weldenegodguad, Juha Kantanen, Jaana Peippo, Kisun Pokharel

**Affiliations:** 1Natural Resources Institute Finland (Luke), 31600 Jokioinen, Finland; 2NordGen—Nordic Genetic Resource Centre, 1432 Ås, Norway

**Keywords:** adaptation, fat tissue, gender, mRNA

## Abstract

Reindeer are native to harsh northern Eurasian environments which are characterized by long and cold winters, short summers, and limited pasture vegetation. Adipose tissues play a significant role in these animals by modulating energy metabolism, immunity, and reproduction. Here, we have investigated the transcriptome profiles of metacarpal, perirenal, and prescapular adipose tissues in Even reindeer and searched for genes that were differentially expressed in male and female individuals. A total of 15,551 genes were expressed, where the transcriptome profile of metacarpal adipose tissue was found to be distinct from that of perirenal and prescapular adipose tissues. Interestingly, 10 genes, including *PRDM9*, which is known to have an important role in adaptation and speciation in reindeer, were always upregulated in all three tissues of male reindeer.

## 1. Introduction

Adipose tissues serve a variety of functions, including storing lipids for energy metabolism and secreting signaling molecules and hormones that play an important role in metabolism, immunity, and reproduction [1,2,3,4,5,6]. The most prevalent functional components of adipose tissues—white and brown adipocytes—store energy in the form of lipids and regulate and respond to the endocrine processes involved in thermohomeostasis and energy balance. The amount and distribution of adipose tissue in mammals can differ significantly between males and females [6,7,8,9,10,11]. Previous studies in humans have shown that sex and sex hormones influence adipose tissue in terms of adipocyte development, adipogenesis, gene expression profiles regulating insulin resistance and lipolysis, and inflammatory response [6,12,13,14,15]. Although sex hormones and steroids are involved in fundamental sex differences in adipose tissue distribution and inflammation, sex chromosomes also determine adipose tissue distribution, along with metabolic and inflammatory responses [6,9,16].

Gender differences in fat distribution, as well as their links to metabolic health, have been widely investigated in clinical and epidemiological studies [6,9,12,15,17]. However, sex-specific differences in studies regarding adipose transcriptome profiling and regulation are relatively scarce. Gender differences in adipose tissue gene expression have mainly been studied in humans and mice [17,18,19,20,21,22,23,24]. Building on these prior studies, we have recently investigated transcriptome profiles of three adipose tissues from different anatomical depots (metacarpal, perirenal and prescapular) in Finnish and Even reindeer (from the Sakha Republic, Russia) during the spring and winter to obtain better insights into how animals in the arctic adapt and survive during harsh conditions [25]. The study found genes associated with energy metabolism and the immune system that responded differently to seasonal and biogeographical changes.

In the present study, for the first time, we have investigated gene expression differences between male and female semi-domestic reindeer in three adipose tissues collected from different anatomical locations (metacarpal, perirenal, and prescapular). The samples were collected from Even (Sakha Republic, Yakutia, Russia) reindeer breed (or population) which have adapted to the harsh Yakutian environments. In this region, the annual temperature may fluctuate from −60 °C up to +30 °C, winters are long, summers are short, and the reindeer feed on natural pastures year-round.

## 2. Materials and Methods

### 2.1. Sample Collection for Transcriptome Analysis

The current study includes RNA-seq of 18 tissue samples from six reindeer individuals (three adult females and three adult males) that were randomly collected at slaughter in the Eveno-Bytantay District, Sakha (Yakutia), the Russian Federation in the winter (November–December) (Table 1 and Appendix A). The male samples were recently included in the study by Weldenegodguad et al. [25]. Perirenal samples were taken from the adipose tissue around the kidneys, prescapular samples from the adipose tissue located beneath the cervical muscles in front of the scapula, and metacarpal samples from the bone marrow in the diaphysis of the metacarpal bone (left front leg). For convenience, throughout the text, the sample groups are abbreviated using gender (M, F) and tissue type (P, S, M). For example, M-M represents the metacarpal tissue of male reindeer. The samples were stored in RNAlater^®^ Solution (Ambion/QIAGEN, Valencia, CA, USA). The animals had been grazing on natural pastures throughout the year before the sampling. The animals were exposed to seasonal ambient temperatures and photoperiods. The mean daily temperature in Northern Sakha varied between −13 °C and −24 °C before sampling (6.5 h of daylight, 17.5 h of dark). All protocols and sample collections were performed in accordance with the legislation approved by the Russian authorization board (FS/UVN 03/163733/07.04.2016s).

### 2.2. RNA Extraction, Library Preparation, and Sequencing

RNA extraction, library preparation, and sequencing were performed at the Finnish Functional Genomic Centre (FFGC), Turku, Finland. Total RNA was extracted from adipose tissues (ca < 30 mg/sample) using the Qiagen AllPrep DNA/RNA/miRNA kit, here, according to the manufacturer’s protocol. The quality of the RNA samples was ensured with an Agilent Bioanalyzer 2100 (Agilent Technologies, Waldbronn, Germany), and the concentration of each sample was measured with a Nanodrop ND-2000 (Thermo Scientific; Wilmington, NC, USA) and a Qbit(R) Fluorometric Quantification, Life Technologies. All samples revealed an RNA integrity number (RIN) above 7.5.

Library preparation was done according to Illumina TruSeq^®^ Stranded mRNA Sample Preparation Guide (part #15031047). Unique Illumina TruSeq indexing adapters were ligated to each sample to pool several samples later in one flow cell lane. Library quality was inferred using an Advanced Analytical Fragment Analyser and concentration with a Qubit fluorometer, and only good-quality libraries were sequenced.

The samples were normalized and pooled for automated cluster preparation, which was carried out with Illumina cBot station. All libraries were combined in one pool and run on the Illumina HiSeq 3000 platform. Paired-end sequencing with a 2 × 75 bp read length was used. Base calling and adapter trimming were performed using Illumina’s standard bcl2fastq (v2.20) software.

### 2.3. Bioinformatic Analyses

The overall quality of the raw RNA-seq reads in fastq and aligned reads in BAM format were examined using FastQC software v0.11.7 [26]. The FastQC reports were summarized using MultiQC v1.7 [27]. High-quality RNA-seq reads for each sample were mapped against the reindeer draft assembly [28] using Spliced Transcripts Alignment to a Reference (STAR) (version 2.6.0a) [29], here, with the default parameters. Next, we generated read counts from the aligned files using the featureCounts software (version 1.6.1) from the Subread package [30] to assign the reads to genes. The GTF format annotation file associated with the reindeer draft assembly was used for gene coordinate information.

To examine the shared and uniquely expressed genes across the three adipose tissues, we used the cpm function from the edgeR library [31] to generate count-per-million (CPM) values. Lowly expressed transcripts with CPM < 0.5 were discarded.

Adipose transcriptomes are affected by the type of adipose, as well as by the sex of the animal [32]. We conducted differential gene expression analysis between male and female reindeer for each adipose tissue.

Raw read counts were processed using the R Bioconductor package DESeq2 [33] to perform differential gene expression and related quality control analyses. Prior to running DESeq2, lowly expressed (rowSums < 1) genes were discarded. Raw gene expression counts were normalized for differences in library size and sequencing depth using DESeq2 to enable gene expression comparisons across samples. We performed principal component analysis (PCA) to assess sample similarity using the variance stabilizing transformation (VST) method. In the current study, we used fold change and false discovery rate (FDR) filtering criteria to identify significantly differentially expressed genes (DEGs). To screen for significant DEGs, we set the absolute value of log2 fold change (LFC) to be greater than or equal to 1.5 (|log2FoldChange| > 1.5) and an adjusted *p*-value of 0.05 (p_adj_ < 0.05). The Benjamini–Hochberg FDR method was used to calculate the adjusted *p*-values.

To gain insights into the biological functions and relevance of the identified DEGs, a functional enrichment analysis was conducted using AgriGO v2.0 [34]. In the AgriGO analysis toolkit, to detect the significantly enriched GO terms, default parameters were used in the ‘Advanced options’: Fisher as the statistical test method, Yekutieli for multiple test correction at a significance level threshold 0.05 (FDR < 0.05) and a minimum number of five mapping entries. In this analysis, the GO annotation file from the de novo assembled reindeer genome [28] was used as a background reference. Furthermore, to explore the biological pathways associated with the DEGs, we performed Kyoto Encyclopaedia of Genes and Genomes (KEGG) pathway analysis using the GAGE [35] Bioconductor package. The significantly enriched pathways were identified based on the *q*-values obtained from Fisher’s exact test (*q*-value < 0.1).

## 3. Results and Discussion

### 3.1. RNA Sequencing and Mapping

A total of 53 gigabases (Gb) of RNA-seq data were generated from 18 adipose tissue samples collected from six Even reindeer. Because the adapters were trimmed automatically and the Phred quality scores of the reads from all samples were greater than 30, we did not perform further trimming and quality filtering. The number of reads per sample ranged from 30.0 million (M) (4.5 Gb) to 39.3 M (5.8 Gb), with a mean high-quality 33.6 M and 2 × 75 bp pair-ended reads per sample (Appendix A). The proportion of reads mapped to the reindeer reference genome ranged from 88.7% to 91.3%, with, on average, >90% of the reads from each sample uniquely mapped to the reindeer draft genome assembly (Appendix A). In general, the mapping statistics indicated good quality for downstream gene expression analyses and were similar to other livestock transcriptome studies [36]. The raw sequence reads in the compressed fastq format (fastq.gz) that were analyzed in the current study are part of a larger study [25]. The data have been deposited to the European Nucleotide Archive (ENA) and are publicly available under accession PRJEB44094.

### 3.2. Gene Expression Overview

A total of 15,551 genes were expressed (cpm > 0.5 in at least two samples) in the 18 samples (Appendix A), representing approximately 56.6% of the 27,332 reindeer genes reported in the draft reindeer genome assembly annotation file [28]. The highest number of genes were expressed in metacarpal adipose tissue (*n* = 14,481), followed by perirenal (*n* = 14,085) and prescapular (*n* = 13,954) adipose tissues. In both male and female reindeer, >12,800 genes were commonly expressed between the adipose tissues (Figure 1A). In both male and female reindeer, the highest number of genes were expressed in the metacarpal adipose tissue, which also displayed the highest number of uniquely expressed genes. To assess the expression similarity among the samples, we performed PCA based on the top 500 most variable genes (Figure 1B). We identified that the gene expression profiles of the metacarpal tissue in both the female and male samples were distinct. These observations are in good agreement with our recent reindeer transcriptome study [25]. The number of genes expressed in the metacarpal tissues was the highest, and in the PCA plot, the metacarpal samples formed a separate cluster from the other tissue samples. The important physiological role of the metacarpal adipose tissue, for example, in energy metabolism of reindeer, has been discussed by Weldenegodguad et al. (2021) [25] and the references therein.

### 3.3. Uniquely Expressed Genes

We explored genes specific to male and female reindeer (sex-biased genes in the present context) in each adipose tissue by removing the lowly expressed genes based on the mean transcripts per million (TPM < 1). In metacarpal adipose tissue, out of 825 genes, 290 and 282 genes were uniquely expressed in male and female reindeer, respectively (Figure 2a). Similarly, perirenal adipose tissue revealed 46 and 28 uniquely expressed genes in male and female reindeer (Figure 2b), respectively, and 31 and 35 genes in the prescapular for male and female reindeer, respectively (Figure 2c).

To further explore the specific genes expressed in male and female reindeer in each adipose tissue, we selected highly expressed genes based on the TPM values. In male reindeer metacarpal adipose tissue, genes such as *IBSP*, *IGLC7*, *EDIL3*, *IGFBP2*, *TRIM63*, *LDB2*, *CHST6*, *DAZAP2*, *UBE2E2*, and *ACAN* were found to be highly expressed genes (TPM > 17.12), whereas *CCDC190*, *CCL26*, *ENHO*, *ISM1*, *CPZ*, *OTOS*, *PGF*, *GNAO1*, and *VMO1* and *MT-CYB* were among the highly expressed genes (TPM > 11.72) in female reindeer. In general, a group of C-C chemokines and collagens were uniquely expressed in the male samples, whereas ribosomal proteins were specific to female metacarpal samples. Sex-biased genes have multifunctional roles of which we provide examples here. The collagens (*COL24A1*, *COL4A3*, *COL6A3*, and *COL6A5*) and *IBSP* are major structural proteins of the bone matrix and are associated with bone mineral density formation. Being male-specific, these genes might contribute to larger bone elements in males compared with females [37,38]. Similarly, immune-related genes such as chemokines (*CCR5*, *CCR6* and *CCR7*), *IGLC7*, *IGKV2-30*, *IL2RA*, and *IGLV1-51* were uniquely expressed in male metacarpal tissues. Moreover, the genes related to spermatogenesis (*DAZAP2* and *IZUMO2*) were also unique to male samples. On the other hand, ribosomal proteins (*MRPS26*, *RPS15A*, *RPS19*, *RPS20* and *RPS6*) and placenta-related (*PGF* and *PLAC8*) genes were uniquely expressed in female metacarpal genes.

The perirenal-specific genes in male reindeer include several ribosomal proteins (*RPL23A*, *RPL27A*, *RPL39*, *RPLP1*, *RPS11*, *RPS13*, *RPS14*, and *RPS23*), and in female reindeer genes, those specifically expressed in perirenal include Ubiquitin-conjugating enzyme (*UBE2C* and *UBE2QL1*) and Zinc finger protein (*ZNF324* and *ZNF775*). Similarly, the scapular-specific genes in male reindeer include ribosomal proteins (*RPL39*, *RPL7*, and *RPS13*) and immune response-related genes (*IGLV3-19* and *MS4A2*). Ribosomal and zinc finger proteins have essential functions for reindeer adaptation in general, as reported previously [28]. Go enrichment analyses did not reveal any significant GO terms.

### 3.4. Differential Gene Expression Analyses

Our comparison of the gene expression profiles between female and male reindeer revealed a total of 327 significant DEGs in the three adipose tissues (Table 2, Appendix A).

We identified a total of 225 significant DEGs between female and male reindeer in metacarpal tissue (M-F vs. M-M) (Figure 3A, Table 2 and Appendix A). Of 225 significant DEGs, 78 genes were upregulated and 147 were downregulated in M-F (Table 2 and Appendix A). The upregulated genes in the female samples included *ANGPTL1*, *CX3CR1*, *CYP2B11*, *CYP2F3*, *CYP4B1*, *FABP6*, *ELOVL7*, *IGHV3-6*, *SLC19A3*, *SLC35F1*, and *SLC6A17* genes, while genes such as *BMP1*, *BMP*, *IBSP*, *EFS*, *CCL19*, *IGDCC4*, *IL17RB*, *NAV3*, *NCAM1*, *NRL*, *OLFML2B*, *OLFML3*, *PRDM9*, *SCUBE1*, and *FLT4* were upregulated in the male samples.

In perirenal tissue, a total of 104 significant DEGs were detected, of which 54 genes were upregulated in females, and the rest were upregulated in males (Figure 3B, Table 2 and Appendix A). The upregulated genes in female samples included *ABCG1*, *ACSL6*, *SLC14A1*, *SLC16A2*, *SLC4A10*, *SLC9A2*, *ZNF219*, *PTER*, *PLA2G5*, *ETNPPL*, *PLCD3*, *S1PR3*, and *S1PR3*. The genes *ATP5H*, *CCL3*, *COX7A1*, *CXCL9*, *NMB*, *NRL*, *PRDM9*, *SLC19A1*, *SLC22A5*, *ZFX*, and *ZRSR2* were upregulated in the male samples.

Furthermore, a total of 49 significant DEGs were detected between female and male reindeer in prescapular tissue (S-F vs. S-M), of which 27 and 22 genes were upregulated and downregulated in S-F, respectively (Figure 3C, Table 2 and Appendix A). *GYS2*, *ARHGEF5*, *ABCC4*, *SUMO1*, *RPL39*, *SLC4A10*, and *ACSL6* are examples of upregulated genes in female samples, and genes such as *TXLNG*, *DDX3Y*, *USP9X*, *EIF2S3X*, *PRDM9*, *KDM6A*, *ZRSR2*, and *UCP1* examples of downregulated genes.

The gene expression analysis of gender differences in adipose tissues indicated that genes associated with fatty acid metabolism and male sterility were upregulated in female and male reindeer, respectively. The upregulated genes associated with fatty acid metabolism in females include *ELOVL7* and *FABP6* in metacarpal, *ABCG1*, *ACSL6*, *PLCD3*, and *PLA2G5* in perirenal and *ACSL6* and *PLA2R1* in prescapular tissue. This result is consistent with previous studies in humans and mice [39,40] that have suggested that females show higher levels of fat deposition than males. Interestingly, the upregulated genes in male reindeer revealed 10 shared genes among the three tissues, such as *UBA1*, *TXLNG*, *PRDM9*, *EIF2S3X*, *NRL*, *DDX3Y*, *KDM6A*, *ZRSR2*, *USP9X*, and *ZFX*. Six of these genes, *PRDM9* [41,42], *UBA1* [43], *EIF2S3X* [44,45], *DDX3Y* [46,47,48], *ZRSR2* [49], and *USP9X* [50] have been shown to be associated with male sterility. In our recent study, as well as by others, *PRDM9* has an important role in reindeer adaptation and survival in the subarctic regions [28]. *PRDM9* is one of the rapidly evolving genes in reindeer and is involved in recombination events and epigenetic modifications [51]. Moreover, recent studies on mice and rats showed that *PRDM9* has important roles in spermatogenesis and male fertility [41,52]. Similarly, other upregulated genes in male samples have diverse functions and may be linked to male infertility or spermatogenic failure (*DDX3Y*), mRNA splicing (*ZRSR2*), lipid metabolism (*EIF2S3X*), circadian rhythm (*USP9X*), cell cycle regulation (*TXLNG*), and photoreceptor development/function (*NRL*).

Moreover, it should be noted that the reindeer reference genome lacked gene annotations for the Y chromosome and that many of the Y chromosome-specific RNA-seq reads may have aligned to the paralog genes of the X chromosome.

### 3.5. GO Enrichment Analyses

From the list of 225 significantly DEGs identified in the M-F vs. M-M comparison (Appendix A), 59 genes did not have GO annotations. The GO analysis results of the upregulated genes in male metacarpal tissue (*n* = 108 with GO annotation) revealed 31 significantly enriched GO terms, whereas those upregulated in female samples (*n* = 58 with GO annotation) resulted only one GO term ‘Oxidoreductase activity, acting on paired donors, with incorporation or reduction of molecular oxygen’ (Appendix A). The upregulated genes in males were significantly enriched in 18 biological processes, 10 molecular functions and 3 cellular component categories (Appendix A). Out of the 18 biological processes, 9 GO terms were associated with developmental processes, such as ‘circulatory system development’, ‘cardiovascular system development’, ‘blood vessel development’, ‘vasculature development’, and ‘anatomical structure development’ (Figure 4). Moreover, the biological process GO terms, ‘cell adhesion’, ‘angiogenesis’, ‘blood vessel morphogenesis’, and ‘anatomical structure formation involved in morphogenesis’ were also significantly enriched in the upregulated DEGs in male metacarpal tissue.

From the list of 104 significant DEGs identified in the P-F vs. P-M comparison (Appendix A), 15 lacked a GO annotation. The GO analysis result of the upregulated (*n* = 48 with GO annotation) DEGs in female perirenal tissue revealed six significantly represented cellular component categories, including “plasma membrane”, “cell periphery”, “plasma membrane part,” and “integral component of membrane” (Appendix A). No significantly represented GO terms were associated with upregulated DEGs in male (*n* = 41 with GO annotations).

From the list of 49 significant DEGs identified in the S-F vs. S-M comparison (Appendix A), seven genes lacked a GO annotation. The GO analysis result of the upregulated (*n* = 23 with GO annotation) and downregulated (*n* = 19 with GO annotation) DEGs in female prescapular tissue revealed three significant enriched molecular function categories and no significantly enriched GO terms, respectively. The significantly enriched GO terms in downregulated DEGs in female reindeer prescapular tissue included ‘cation binding’, ‘ion binding’ and ‘metal ion binding’, indicating their role in the acquisition of mineral nutrients: iron, zinc, and calcium.

### 3.6. KEGG Pathway Analyses

KEGG pathway analysis revealed no statistically significant enriched KEGG pathways between M-F and M-M. ‘Ribosome’ was the only KEGG pathway associated with DEGs from the perirenal and prescapular adipose tissues. Interestingly, in both tissues, the ‘ribosome’ pathway was downregulated in female reindeer.

## 4. Conclusions

In the present study, we have investigated male versus female differences in gene expressions in three adipose tissues in the Even reindeer population originating from Northern Sakha (Yakutia), which is one of the coldest geographic regions where reindeer herding has held a long tradition in people’s livelihoods. To the best of our knowledge, this is the first transcriptome study comparing gender differences in reindeer. In general, sex differences in adipose tissue transcriptomes have mainly been examined in humans and mice. One of the key observations made in the present study was that metacarpal adipose tissue displayed distinct gene expression profiles compared with the other two tissues. In addition, we found that the metacarpal tissue showed the highest number of gender-specific genes expressed only in the female and male samples. Among the list of DEGs, *PRDM9* was specifically upregulated in all three adipose tissues of males, indicating its significant functional role in male fertility and the evolution of reindeer in general.

## Figures and Tables

**Figure 1 genes-13-01645-f001:**
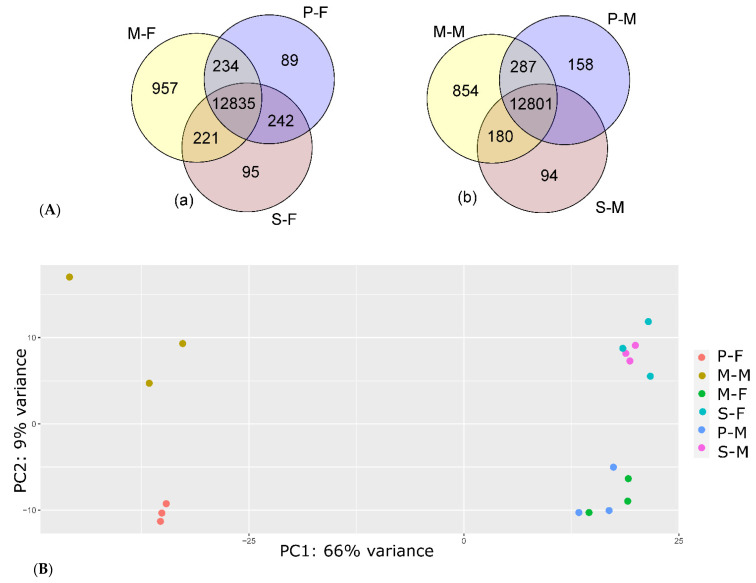
Sample relatedness. (**A**) Venn diagram showing overlap of expressed genes (CPM ≥ 0.5 for at least two samples) among the tissues in male (**a**) and female (**b**) reindeer. (**B**) PCA plots of the analysed samples based on expression profiles.

**Figure 2 genes-13-01645-f002:**
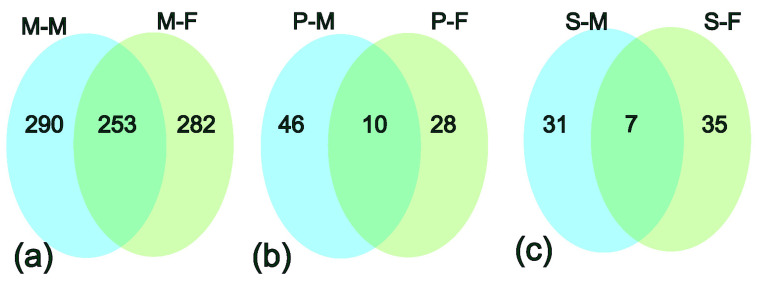
Distribution of uniquely expressed genes in each adipose tissue in male and female reindeer. (**a**) Shared and uniquely expressed genes in the metacarpal adipose tissue of male (M-M) and female (M-F) samples, (**b**) shared and uniquely expressed genes in the perirenal tissue of male (P-M) and female (P-F) samples, and (**c**) shared and uniquely expressed genes in the scapular tissue of male (S-M) and female (S-F) samples.

**Figure 3 genes-13-01645-f003:**
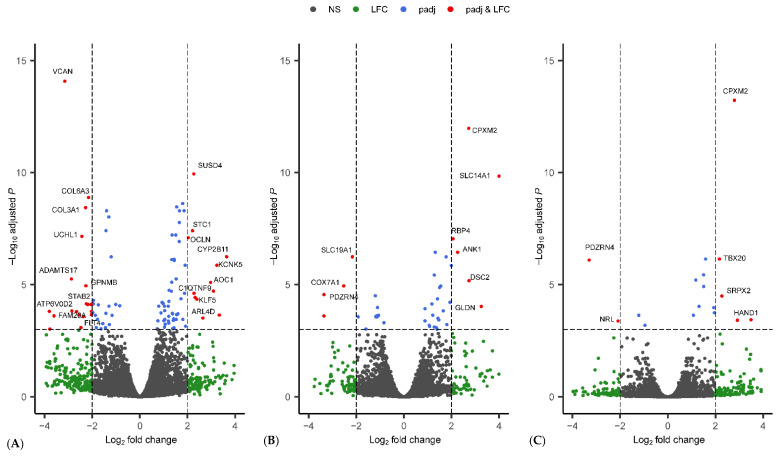
Volcano plot of genes that were expressed in the (**A**) metacarpal adipose tissue, (**B**) perirenal adipose tissue, and (**C**) prescapular adipose tissue. Significantly differentially expressed genes are marked with their gene names. For the complete list of differentially expressed genes, please see Appendix A. In figure, NS = non-significant; LFC = absolute Log_2_ fold change greater than 2; padj = adjusted *p* value less than 0.001; and padj and LFC represent genes that fulfil the criteria of threshold for LFC and padj.

**Figure 4 genes-13-01645-f004:**
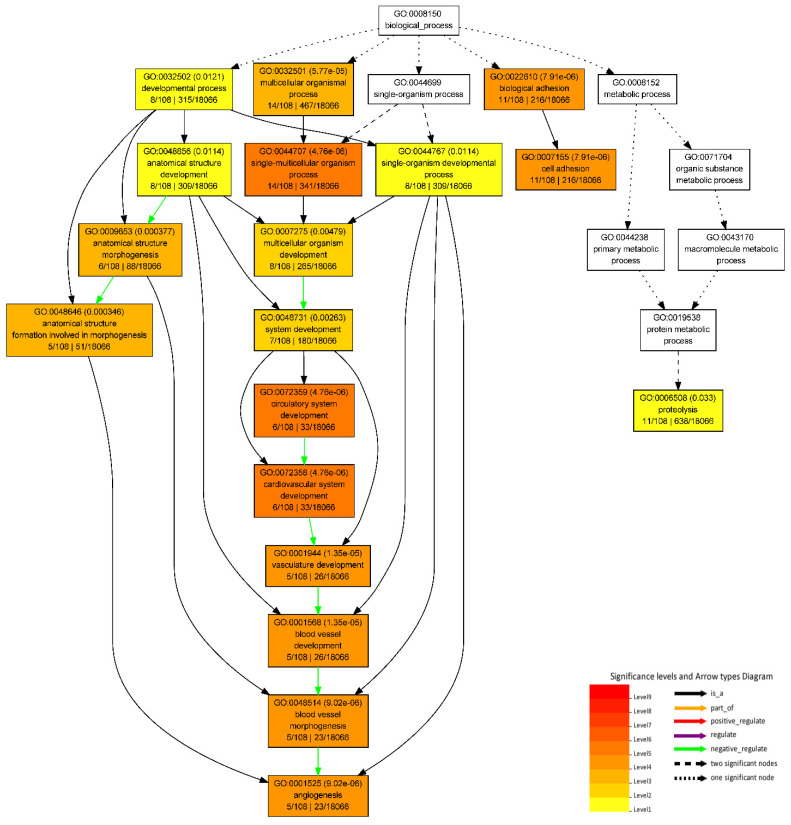
GO hierarchical graph of 18 biological processes associated with upregulated genes in the metacarpal adipose tissue of male samples.

**Table 1 genes-13-01645-t001:** Summary of adipose tissue samples.

	Male	Female
Metacarpal	3	3
Perirenal	3	3
Prescapular	3	3

**Table 2 genes-13-01645-t002:** Summary of the DEGs from male (-M) and female (-F) comparisons in metacarpal (M), perirenal (P) and prescapular (S) adipose tissues of Even (E) reindeer.

Comparison	Total	Upregulated	Downregulated
M-F vs. M-M	225	78	147
P-F vs. P-M	104	54	50
S-F vs. S-M	49	27	22

## Data Availability

Raw sequence reads in compressed fastq format (fastq.gz) have been deposited to the European Nucleotide Archive (ENA) and are publicly available under accession PRJEB44094.

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
