# Peer review of "Differences in Adipose Gene Expression Profiles between Male and Female Even Reindeer (*Rangifer tarandus*) in Sakha (Yakutia)"

_genes, 2022, doi:10.3390/genes13091645_

Round 1

Reviewer 1 Report

Comments to the author:

In this study, the authors are the first to systematically compare the transcriptome difference of three different adipose tissue depots (metacarpal, perirenal, and prescapular) between female and male reindeer. This research contributes to a better understanding of how animals adapt and survive in harsh environmental conditions.

There are some problems in the article, and the suggestions are as follows:

1. What kind of adipocytes (white or brown adipocytes) are the main components of metacarpal adipose tissue?

2. In result 3.4, to visualize differentially expressed genes, I recommend plotting heatmaps of significant DEGs mentioned in the text.

3. In result 3.5, to visualize significantly enriched GO terms, I recommend plotting the GO functional enrichment graphs instead of a table.

4. In result 3.6, the title should be “KEGG enrichment analyses”.

5. I recommend that the Results and Discussion sections be separated.

6. In the Results section, please increase the resolution of all figures in the article to better visualize these results.

7. There are some verbal errors throughout the manuscript.

Author Response

In this study, the authors are the first to systematically compare the transcriptome difference of three different adipose tissue depots (metacarpal, perirenal, and prescapular) between female and male reindeer. This research contributes to a better understanding of how animals adapt and survive in harsh environmental conditions.

Author response: Thank you very much for reviewing our work. We have tried our best addressing your comments and concerns regarding our manuscript.

There are some problems in the article, and the suggestions are as follows:

  1. What kind of adipocytes (white or brown adipocytes) are the main components of metacarpal adipose tissue?

AR: Thank you for this suggestion. Metacarpal adipose tissue comprises of both the white and brown adipocytes. We provided more information about the tissue types in our recently published paper that has been cited several times here. Therefore, we decided not to go into the details.

  1. In result 3.4, to visualize differentially expressed genes, I recommend plotting heatmaps of significant DEGs mentioned in the text.

AR: Thank you for this recommendation. We realized that the DEGs are not well represented visually and decided to prepare volcano plot (Fig 3). While we also prepared a heatmap plot based on your suggestion, volcano plot appeared to be more informative.

  1. In result 3.5, to visualize significantly enriched GO terms, I recommend plotting the GO functional enrichment graphs instead of a table.

AR: Thank you for this excellent suggestion. We have added a GO hierarchical plot (please see Fig 4) of all biological processes associated with 108 genes that were upregulated in male samples of metacarpal adipose tissues.

  1. In result 3.6, the title should be “KEGG enrichment analyses”.

AR: Thank you for pointing out the typo. We have changed the title “KEGG pathway analyses”.

  1. I recommend that the Results and Discussion sections be separated.

AR: Thank you for this suggestion. We had originally prepared the manuscript by separating results and discussion sections. However, we realized that several of the results would need to be repeated to make the discussion section reader friendly. We then decided to merge and to our mind, the current format suits the content our manuscript.

  1. In the Results section, please increase the resolution of all figures in the article to better visualize these results.

AR: Thank you for this suggestion. We have not only improved the resolution of the figures but also readability. All the figures are submitted separately as PDF files.

7. There are some verbal errors throughout the manuscript.

AR: This manuscript went through English language editing service. We have also tried our best to correct all the grammatical and verbal errors.

Reviewer 2 Report

The authors planned and executed the research very well. The results were nicely presented, however, there are some concerns in the manuscript which are presented as below.

It is recommended that the authors reorganize the statements in the abstract section so that the reader can better understand the authors' work.

It is suggested that the author should add the statistical results of identifying differentially expressed genes to the body of the article in the form of pictures.

Please check that the relevant references are fully and correctly cited in the Bioinformaticanalyses section. And whether the correct statistical method for analysis?

In line 228 of the article, there are 2 PLCD3.

It is suggested that the author should add the important differentially expressed gene information in the form of table to the text of the article.

It is suggested that the author should add the results of GO and KEGG analysis to the body of the article in the form of pictures.

Author Response

The authors planned and executed the research very well. The results were nicely presented, however, there are some concerns in the manuscript which are presented as below.

Author response: Thank you very much for reviewing our work. We have tried our best addressing your comments and concerns regarding our manuscript.

It is recommended that the authors reorganize the statements in the abstract section so that the reader can better understand the authors' work.

AR: Thank you for the recommendation. We have now modified the abstract section.
Reindeer are native to harsh northern Eurasian environments which is characterized by long, cold winters, short summers, and limited pasture vegetation. Adipose tissues play significant role in these animals by modulating the energy metabolism, immunity, and reproduction. Here we have investigated the transcriptome profiles of metacarpal, perirenal and prescapular adipose tissues in Even reindeer and searched for genes that were differentially expressed in male and female individuals. A total of 15,551 genes were expressed, where the transcriptome profile of metacarpal adipose tissue was found to be distinct to that of perirenal and prescapular adipose tissues. Interestingly, 10 genes, including PRDM9 which is known to have important role in adaptation and speciation in reindeer, were always upregulated in all three tissues of male reindeer.

It is suggested that the author should add the statistical results of identifying differentially expressed genes to the body of the article in the form of pictures.

AR: Thank you for this suggestion. We have added volcano plots to represent significantly differentially expressed genes (please see Fig 3).

Please check that the relevant references are fully and correctly cited in the Bioinformatic analyses section. And whether the correct statistical method for analysis?

AR: Thank you for the reminder. We can make sure that all bioinformatics tools and references are properly cited. We have used already established and standard statistical methods for gene expression analysis.

In line 228 of the article, there are 2 PLCD3.

AR: Thank you. We now removed one of the PLCD3.

It is suggested that the author should add the important differentially expressed gene information in the form of table to the text of the article.

AR: Thank you for the suggestion. Instead of table, we highlighted important differentially expressed genes in the volcano plot that we believe provide more information to readers (please see Fig 3).

It is suggested that the author should add the results of GO and KEGG analysis to the body of the article in the form of pictures.

AR: Thank you for this suggestion. We now added Fig 4 which represents hierarchical plot of all biological processes associated with genes that were upregulated in male samples of metacarpal adipose tissue. Other categories either did not result into GO terms or were not meaningful in the context of our study, and thus have been included as supplementary tables.